# Pleiotropic Signaling by Reactive Oxygen Species Concerted with Dietary Phytochemicals and Microbial-Derived Metabolites as Potent Therapeutic Regulators of the Tumor Microenvironment

**DOI:** 10.3390/antiox12051056

**Published:** 2023-05-06

**Authors:** Toshiyuki Murai, Satoru Matsuda

**Affiliations:** 1Graduate School of Medicine, Osaka University, 2-2 Yamada-oka, Suita 565-0871, Japan; 2Department of Food Science and Nutrition, Nara Women’s University, Kita-Uoya Nishimachi, Nara 630-8506, Japan

**Keywords:** redox signaling, tumor microenvironment, cancer stem cells, epithelial-to-mesenchymal transition, phosphoinositide 3-kinase, PI3K/AKT

## Abstract

The excessive generation of reactive oxygen species (ROS) plays a pivotal role in the pathogenesis of diseases. ROS are central to cellular redox regulation and act as second messengers to activate redox-sensitive signals. Recent studies have revealed that certain sources of ROS can be beneficial or harmful to human health. Considering the essential and pleiotropic roles of ROS in basic physiological functions, future therapeutics should be designed to modulate the redox state. Dietary phytochemicals, microbiota, and metabolites derived from them can be expected to be developed as drugs to prevent or treat disorders in the tumor microenvironment.

## 1. Introduction

Reactive oxygen species (ROS) is a term used for a family of reactive species derived from molecular oxygen. ROS are continuously generated and scavenged in the cells of all aerobic organisms. The term ROS is useful for a global description, but the name of the specific chemical species referred to must be used as and when required [1]. Superoxide anion (O_2_^−•^), hydroxyl (OH^•^), alkoxyl (RO^•^), and peroxyl (ROO^•^) are the major free-radical ROS, while hydrogen peroxide (H_2_O_2_), ozone (O_3_), singlet molecular oxygen (O_2_^1^Δ_g_), electronically excited carbonyls (RCO), and organic hydroperoxide (ROOH) are the major nonfree-radical forms of ROS (Table 1).

The excessive generation of ROS plays a pivotal role in the pathogenesis of many diseases [2]. Improper regulation of ROS levels contributes to the pathologies of many maladies including inflammation, cancer, and neurodegeneration [3,4,5]. Increased ROS production adversely affects the mitochondrial electron transport chain, the site of oxidative phosphorylation in eukaryotes where a series of electron transporters are embedded in the inner mitochondrial membrane, and eventually leads to apoptosis [2,6]. ROS are considered harmful to cell components and cause DNA damage [7,8]. Oxidative damage in cells is mainly attributed to excess ROS production. ROS produce metabolic intermediates that are involved in various signaling pathways. The inhibition of the electron transport chain within the mitochondrion results in the generation of ROS, which finally leads to severe injury; as a result, the dysfunction in the mitochondrion induced by cisplatin could be attributed to the increased concentration of ROS resulting in the apoptosis of cells [2,9,10,11,12].

ROS are localized to specific subcellular locations in which a specific metabolite is synthesized, as the surroundings may dictate the targets certain ROS molecules may potentially encounter spatially and temporally. One of the major intracellular mechanisms with ROS generation is the phagosome, a specialized membrane-bound organelle that appears in phagocytic cells including macrophages, dendritic cells, and neutrophils. Belgian biochemist Christian de Duve first discovered that heterogenic intracellular cargoes could be transported to lysosomes for degradation—autophagy, a term he coined himself, during his seminal work on the discovery of lysosomes [13], which led to his being awarded the Nobel Prize in Physiology or Medicine in 1974.

Peroxisomes are single-membrane-bound organelles that harbor enzymes that catalyze many metabolic reactions, including one of the major energy-generating metabolisms. Peroxisomes play a critical role in lipid homeostasis, the β-oxidation pathway for the degradation of long-chain fatty acids, and plasmalogen synthesis. The β-oxidation pathway is a metabolic process consisting of multiple enzymes by which long-chain fatty acids are sequentially degraded concomitantly producing energy. In the β-oxidation pathway, fatty acids primarily go into the cytosol through fatty acid protein transporters, and thereby a coenzyme A (CoA) group is added to the fatty acid. The long-chain coenzyme A can then enter the β-oxidation pathway, resulting in the production of one acetyl-coenzyme A per one cycle of the β-oxidation pathway. Consequently, acetyl-coenzyme A then goes into the tricarboxylic acid cycle (TCA cycle), also known as the Krebs cycle or citric acid cycle. This produces the reduced form of nicotinamide adenine dinucleotide (NADH) and the reduced form of flavin adenine dinucleotide (FADH_2_), which are then utilized in the respiratory chain to generate adenosine 5′-triphosphate (ATP). They contain both ROS-producing and -scavenging enzymes and thus are active in ROS metabolism [14,15,16]. Christian de Duve was also the first who isolated another type of organelle, peroxisome, from the rat liver, and his biochemical investigation led to the discovery of the localization of oxidases that produce H_2_O_2_ and H_2_O_2_-degrading enzymes within peroxisomes [17]. Peroxisomal biogenesis is modulated by a family of peroxisomal proteins called peroxins, which are needed for the construction of the membrane surrounding the peroxisome, transport of the peroxisomal matrix proteins, and division of the peroxisomes. Certain mutations in peroxins lead to metabolic disorders called peroxisomal biogenesis disorders including Zellweger syndrome [18,19]. Other than these reactions within peroxisomes, there are many interactions with extra-peroxisomal sites. For example, catalase is localized in peroxisomes where it regulates oxidative reactions possibly through the suppression of the import of catalase into the peroxisomes by peroxisome transport signals [20]. The peroxisome is an organelle with specialized functions in H_2_O_2_ metabolism, and there is a critical need to improve the understanding of the correlations between intra-peroxisomal and extra-peroxisomal oxidant-based signaling [20]. H_2_O_2_ is recognized as the major ROS to be involved in the redox regulation of biological pathways. Peroxisomes can regulate the amount of reactive species including ROS, reactive nitrogen species (RNS), carbonyl reactive species (CRS), and sulfur reactive species (SRS), and their interactions with and effects on target molecules which contribute to the redox state and ROS homeostasis enable the regulation of signaling networks related to peroxisome-dependent H_2_O_2_ [21,22,23].

## 2. Dietary Phytochemicals Regulate ROS Signaling

Plants synthesize a wide variety of chemical compounds known as secondary metabolites, such as polyphenols and flavonoids, which are essential to their physiology and growth. There is a substantial body of research that has explored the health benefits of these phytochemicals when plants are included in the diet as many of them are likely to have pleiotropic effects including those through ROS regulation, while some effects may be unknown or require further investigation [24,25,26,27] (Figure 1). Dietary polyphenols include the families of flavonoids, stilbenes, and chalcones in addition to nonflavonoid polyphenol compounds [28]. Flavonoids contain a common carbon skeleton structure of diphenyl propane in which two benzene rings connect by a linear three-carbon chain (Figure 1). The main flavonoid sources are fruits, vegetables, and tea. Among the fruits, berries, cherries, plums, and apples are the richest in flavonoids; in contrast, flavonoids are less abundant in tropical fruits [29,30]. Flavonoids are potent antioxidants that protect plants from unfavorable environmental conditions [30]. The hydroxyl groups in flavonols are responsible for their biological activities. These hydroxyl groups are capable of readily donating hydrogen electrons to stabilize a radical species [31].

Quercetin, 2-(3,4-dihydroxyphenyl)-3,5,7-trihydroxychromen-4-one; resveratrol, 5-[(*E*)-2-(4-hydroxyphenyl)ethenyl]benzene-1,3-diol; rutin, quercetin-3-*O*-rutinoside (3′,4′,5,7-tetrahydroxy-3-[α-L-rhamnopyranosyl-(1→6)-β-D-glucopyranosyloxy]flavone; chlorogenic acid ((1*S*,3*R*,4*R*,5*R*)-3-[(*E*)-3-(3,4-dihydroxyphenyl)prop-2-enoyl]oxy-1,4,5-trihydroxycyclohexane-1-carboxylic acid; caffeic acid, (*E*)-3-(3,4-dihydroxyphenyl)prop-2-enoic acid; ferulic acid, (2*E*)-3-(4-hydroxy-3-methoxyphenyl)prop-2-enoic acid.

Quercetin (2-(3,4-dihydroxyphenyl)-3,5,7-trihydroxychromen-4-one) (Figure 1) is a major plant flavonol from the flavonoid group of polyphenols with a bitter flavor and is found in many fruits, vegetables, leaves, and seeds such as capers and red onions. Quercetin has scavenging activity against hydrogen peroxide (H_2_O_2_), hydroxyl radical (OH^•^), and superoxide anion (O_2_^−•^) and exhibits neuroprotective and anti-inflammatory activities. The neuroprotective effects of quercetin are mainly attributed to signaling activities via the nuclear factor-erythroid 2 p45-related factor 2 (Nrf2), Jun N-terminal kinase (JNK), protein kinase C (PKC), mitogen-activated protein kinase (MAPK), and phosphoinositide 3-kinase (PI3K)/protein kinase B (AKT) pathways [32]. Quercetin attenuates inflammation through the NF-κB and Nrf2 pathways [33].

Resveratrol, *trans*-3,4′,-5-trihydroxystilebene, (5-[(*E*)-2-(4-hydroxyphenyl)ethenyl]benzene-1,3-diol) (Figure 1) is an efficient ROS scavenger and exhibits a protective effect against lipid peroxidation in plasma membranes and DNA damage caused by ROS [34]. Resveratrol is rich in the skin of fruits such as grapes, blueberries, and raspberries, and the uptake of resveratrol by red wine consumption could be behind the so-called French paradox [35]. Resveratrol elicits a wide variety of pharmacological effects including cellular defense, which is attributed in part to resveratrol’s activity as a direct antioxidant [36]. Resveratrol activates sirtuin 1 via the AMP-activated protein kinase pathway and prevents inflammation via the inhibition of the NF-κB pathway [36]. Resveratrol attenuates ROS production and the MAPK and NF-κB signaling pathways, exhibiting an anti-inflammatory function [37]. 

Rutin, quercetin-3-*O*-rutinoside (3′,4′,5,7-tetrahydroxy-3-[α-l-rhamnopyranosyl-(1→6)-β-d-glucopyranosyloxy]flavone) is a common dietary flavonoid glycoside found in many plants, including buckwheat and asparagus, exhibiting pharmacological activities including anti-oxidation [38]. Rutin provides a protection effect against neurotoxicity via the activation of the PI3K/AKT/glycogen synthase kinase 3β (GSK-3β) pathway by regulating phosphorylation by scavenging free-radical generation [39]. 

While many reports focused on the antioxidant properties of flavonoids, recent research suggests an emerging view that flavonoids and their metabolites do not solely act as conventional hydrogen-donating antioxidants but rather may exert modulatory actions via intracellular signaling pathways [40]. In particular, flavonoids and their metabolites have been reported to act as modulators for the PKC pathway in addition to the PI3K/AKT and MAPK signaling cascades, which are likely to affect profound cellular function including gene expression.

One of the major types of phytochemicals are polyphenols, which have attracted the attention of nutritionists and biochemists due to epidemiological studies supporting their beneficial effects on human health [41,42,43,44,45]. Foods such as coffee, tea, cocoa, fruits, seeds, and their oils are rich in nonflavonoid polyphenols. As an example, coffee is a rich source of various secondary metabolites and is abundant with several phenolic compounds such as chlorogenic acid, caffeic acid, lactones, and diterpenes [46]. Chlorogenic acid (3-*O*-caffeoylquinic acid; 3-(3,4-dihydroxycinnamoyl)quinic acid (1*S*,3*R*,4*R*,5*R*)-3-[(*E*)-3-(3,4-dihydroxyphenyl)prop-2-enoyl]oxy-1,4,5-trihydroxycyclohexane-1-carboxylic acid) is a polyphenol and the cinnamate ester of quinic acid and caffeic acid (Figure 1). The beneficial effects of these compounds are attributed in part to their antioxidant activities. Chlorogenic acid exerts its antioxidant effects due to its polyhydroxyl-based structure which consists of several hydroxyl groups that readily react with free radicals to form hydrogen free radicals, which eliminate hydroxyl radicals (OH^•^) and superoxide anions (O_2_^−•^) to exhibit a significant antioxidant activity; therefore, they regulate the activities of the endogenous oxidase system and its associated proteins which prevent oxidative damage to cell organelles, proteins, nucleic acids, and lipids, thereby effecting the pathogenesis of cancer, inflammation, and neurological diseases [47,48]. 

Dietary polyphenols including the chemical compounds described above are found in various fruits, vegetables, and seeds and have been studied for their potential health benefits [49]. Currently, available evidence supports this association, and recent clinical studies regarding the use of polyphenol supplementation, which include the doses, duration of the studies, and the specific health outcomes that were measured, have also been summarized [50]. However, a safety assessment of the dose of polyphenols should be performed to prevent adverse effects [51]. Therefore, it would be helpful to refer to the possible side effects of polyphenols which have been comprehensively summarized in [52].

## 3. ROS as Developmental Signals in Oocytes

ROS signaling may play important roles in embryo development. Ovarian tissue is composed of two components, i.e., the ovarian cortex and the ovarian medulla. The ovarian medulla consists of connective tissues and blood vessels in addition to the fibrous tissues [53]. The interaction between oocytes and their surrounding cells is supported by direct association between them through ECM molecules, signaling proteins such as growth factors, and metabolite molecules [53]. The ovarian micro-environment may regulate the status of oocytes and enhance the aging of oocytes which lead to disorders including infertility [53]. High ROS concentration over physiological concentration might destabilize mitosis-promoting factor (MPF) and reduce certain survival factors which lead to the programmed cell death of oocytes mediated by the mitochondrion [54]. Oocytes are active in metabolism in dormancy and therefore have to keep mitochondrial activity for the generation of essential factors, while the mitochondrion is the main generator of ROS, producing ROS just as side products of oxidative respiratory reactions [55]. While ROS are able to act as signaling molecules, highly concentrated ROS promote DNA mutagenesis and thus are harmful to the cell [55]. Mitochondria of early oocytes in culture lack ROS and have low membrane potential, basal respiration rates, and resistance to rotenone indicating that they have evolved to balance their metabolism to enhance longevity, and hence primordial oocytes in humans and *Xenopus laevis* (frogs) lack any detectable ROS signals [55]. While it is true that primordial oocytes in humans and *Xenopus laevis* lack detectable ROS signals, it is important to note that ROS levels increase as oocytes mature. This can have implications for the quality of the resulting embryos. Early oocytes exhibit greatly reduced levels of mitochondrial complex I [55]. Complex I (respiratory complex I) is the first enzyme of the mitochondrial electron transport chain, a proton-pumping oxidoreductase key to bioenergetic metabolism. Primary sources of ROS occur from the transfer of electrons to molecular oxygen at Complex I. Furthermore, artificial induction of ROS synthesis in eggs caused them to degenerate rapidly, suggesting that they may have poorly developed protective mechanisms against ROS-mediated damage and that they use ROS as signals in a self-defense strategy for ensuring a longer life [56]. 

## 4. ROS Signaling in the Tumor Microenvironment

A tumor is not merely a mass of cancer cells, but rather a heterogeneous assembly of infiltrating and resident host cells, various secreted factors, and the extracellular matrix, forming a tumor microenvironment. The tumor microenvironment is a highly complex and dynamic ensemble of cells of which a variety of immune cells and cancer-associated fibroblasts (CAF) are a major component. Currently, increasing evidence indicates that various kinds of immune cells reside within the tumor microenvironment: innate immune cells including neutrophils, macrophages/dendritic cells, and innate lymphoid cells (ILCs) and acquired immune cells including T- and B-lymphocytes. Neutrophils reconstitute the ECM in the tumor microenvironment by producing matrix metalloproteinases and ROS, and macrophages/dendritic cells modulate the migration of cancer cells within the tumor microenvironment by interacting with them. The details of the subtypes of these tumor-microenvironment-related immune cells have been summarized in [57]. These cells secrete vascular endothelial growth factor (VEGF), epidermal growth factor (EGF), and hepatocyte growth factor (HGF) and the subsequent activation of their receptor tyrosine kinase (RTK) signaling pathways in enhancing ECM degradation, angiogenesis, and metastasis. At the early stage of tumor growth, dynamic interactions occur within the tumor microenvironment, where cancer cells are supported by the other components to survive, locally invade, and form metastatic dissemination. The development of secondary tumors in bodily sites distant from the primary tumor is termed metastasis. Though metastasis accounts for the greatest proportion of cancer-associated deaths, it remains the most complex and least understood aspect of cancer biology. Cancer stem cells were defined as a small subpopulation of cells within a tumor that possess the capacity to self-renew and to initiate the heterogeneous lineages of cancer cells that constitute a tumor. 

Cancer stem cells are identified and can be isolated based on the expression of specific cell-surface proteins that act as molecular biomarkers. The markers most frequently used to identify cancer stem cells in solid tumors are CD44, CD133, CD24, epithelial cell adhesion molecule (EpCAM), interleukin 6 (IL-6) receptor, and leucine-rich repeat-containing G-protein coupled receptor 5 (LGR5). The expression profiles of those markers are typically conserved across the cancer cell types of hematogenous and solid tumors. However, the clinical use of cancer stem-cell-specific biomarkers in solid tumors is relatively limited because most markers expressed in cancer stem cells are also expressed in stem cell populations within normal adult tissues. Moreover, there is accumulating evidence that does not support the conventional cancer stem cell hypothesis [58,59].

Although cancer stem cells have been considered as a relatively very small subpopulation of tumor cells in certain malignancies, relative rarity is not a defining criterion within the consensual cancer stem cell definition: relative rarity is not necessarily a common feature that defines all cancer stem cells that have been identified. A characteristic of cancerous stem cells is their adaptability to the heterogeneous tumor microenvironment which includes low pH, hypoxia, nutritional state [60,61], and acidosis which is a consequence of exacerbated glycolysis and the impaired extrusion of acidic waste products such as lactate [62]. Aberrant metabolic changes in cancer cells such as enhanced glucose metabolism induce the generation of excess protons; hence, cancer cells rely on proton exchangers and transporters to export the protons into the microenvironment, allowing them to survive the hostile environment that they create [63]. Therefore, pH regulators expressed in cancer cells such as carbonic anhydrase are possible therapeutic targets, and inhibitors of carbonic anhydrase such as sulfonamides and coumarins displayed inhibitory effects on breast cancer stem cells [63].

However, the adoption of glycolysis metabolism may be promoted in an unforced manner that can occur even in a microenvironment where oxygen is available and cells have proper mitochondrial function [64]. This metabolic reprogramming forces cancer cells to maintain anabolism while decreasing ROS overproduction, which is highly toxic for them as they have low levels of ROS detoxification enzymes [64]. In addition, glucose, fatty acids, and extracellular catabolites can support metabolism in cancer stem cells by serving as alternative fuel. Since cancer cells utilize fatty acid synthesis for obtaining energy from fatty acid metabolism to support cell growth and proliferation and fatty acid oxidation to produce nicotinamide adenine dinucleotide (NADH) and ATP, various candidate drugs targeting lipid metabolism in cancer stem cells, such as fatty acid synthase inhibitors, have entered clinical investigations in recent years [65].

Although metastasis, which is the development of secondary tumors at sites distant from the primary tumor, accounts for the highest proportion of cancer-associated deaths, it remains the most complex and least understood aspect of cancer biology [66]. According to the cancer stem cell model, to establish a distant metastasis, cancer stem cells must intravasate into the bloodstream or lymphatic system for which they must first lose their epithelial characteristics such as cell–cell interactions by tight junctions and adherens junctions, to exhibit mesenchymal characteristics such as a typical fibroblastic morphology, enhanced migration, and augmented invasion capabilities; this epithelial-to-mesenchymal transition (EMT) is triggered by signals that the cells receive from their microenvironment [67,68,69,70]. The EMT International Association (TEMTIA) has noted that cancer cells may be able to migrate without promoting the EMT, possibly by collective cell migration in a manner that frequently occurs in organismal development. However, it is still unknown whether cancer cells in the primary site are able to conduct all the steps of metastatic dissemination cascades without the transient activation of the EMT to some extent [40]. Despite a continued scientific debate on the involvement of the EMT in cancer progression, it appears to be a major strategy utilized by cancer cells to acquire cancer stem-cell-specific phenotype, making this process an attractive target for developing novel cancer therapies [71]. Cancer cells are thought to accomplish metastasis not merely by the contribution of cancer stem cells or tumor-initiating cells but also by the ability of cells to exert capacity in severely adverse conditions [72]. The small population of disseminated tumor cells that are capable of the initiation of distant tumor growth are called metastasis-initiating cells, or metastatic stem cells [73]. The challenge associated with the development of therapies for the prevention or treatment of metastasis is the plasticity and heterogeneity of cancer stem cell populations.

## 5. ROS Signaling Pathway

Several intracellular signaling pathways play crucial roles in normal stem cell function, among which the core pathway is the PI3K-AKT-PTEN axis. The PI3K/AKT pathway plays a critical role in the survival and proliferation of normal cells under standard physiological conditions. Notably, the PI3K-AKT-PTEN pathway is involved in multiple physiological cellular processes including stem cell renewal: the PI3K-AKT-PTEN axis is one of the major pathways for regulating the maturation of erythroid progenitor cells [74]. In neural stem cells, the endogenous ROS levels are regulated by PI3K-AKT-PTEN signaling [75]. The PI3K/AKT pathway also functions in modulating cell behaviors in a variety of cancers [76]. For example, the development of prostate cancer is often associated with the silencing of PTEN, a multifunctional enzyme that inhibits PI3K/AKT signaling in the cytosol, stabilizes DNA in the nucleus, and acts as a tumor suppressor. Various extracellular molecules, including growth factors and nutrients, can act as regulators of the PI3K-AKT-PTEN signal transduction. EGF, one of the major growth factors in TME, activates the PI3K-AKT-PTEN signaling pathway in cancer cells for their survival and invasion [77]. VEGF also activates PI3K-AKT-PTEN signaling to enhance neovascularization in tumors [78]. PI3K on activation recruits protein kinases including AKT and consecutively transduces the signal to downstream molecules in the pathway [79].

In general, the intracellular signaling networks coordinate cell functions such as cell cycle progression, proliferation, and protection. Mutations in the components of a specific pathway could consequently activate or inhibit the relevant signaling transduction. Mutations in the components of intracellular signaling pathways can lead to dysregulated signaling in many malignancies, which can promote cancer cell proliferation and survival. Hence, PTEN/PI3K/AKT activities in both the nucleus and cytosol should theoretically correlate with the clinical and pathological parameters associated with cancer [79]. Moreover, PI3K/AKT signaling is involved in the induction and expression of a gene encoding a protein conferring multidrug resistance, which is conspicuously associated with cancer therapy antagonism, a major obstacle in the effective chemotherapy of cancer. As the PI3K/AKT signaling pathway is a remarkable molecular target for the designing of novel methods directed at the therapy of cancer and related conditions, many pharmacological inhibitors of the pathway have been developed [80]. Although many chemical compounds are developed to inhibit PI3K and introduced into clinical trials including cancer treatment, a challenge associated with the specificity of PI3K is to distinguish how each PI3K isozyme regulates the physiology in normal conditions [81]. As one example, a study presented isoform-specific PI3K inhibitors [82]. In that study, the most selective PI3K inhibitor is a quinazolinone purine compound named PIK-39, which inhibits a Class I PI3K isozyme p110δ at nanomolar concentration. ROS play a role as second messengers that regulate redox-sensitive signals, such as the stress-activated MAPK, JNK, and p38 MAPKs [83,84]. Therefore, ROS conduct multiple functions: the function of the activation of Ask1 by removing Trx and the function of the promotion of AKT activities. Events triggered via the PI3K pathway entail the activation of NADPH oxidase and production of ROS [85]. ROS signaling is linked to other signaling pathways such as starvation-induced autophagy: a recent report showed that starvation induces the generation of ROS, specifically H_2_O_2_, which is indispensable for autophagy through the oxidation of HsAtg4 protein, thus providing a molecular mechanism underlying the oxidative regulation of the autophagy [86]. Therefore, in addition to the conventional chemotherapeutic approaches, the modulation of oxidative stress may be a novel efficacious modality for cancer treatment [87].

## 6. Gut Microbiota and ROS

Human-body-associated microbes play an important role in modulating health, and an imbalance in the microbial profile has been linked to a wide range of diseases; the associations between the microbiome and host phenotypes have been summarized [88]. For example, the amount of the population of *Bacteroides* sp. is correlated with obesity, diabetes, and Parkinson’s disease [88]. The abundance of certain bacterial species is associated with a healthy metabolic condition and is crucial in designing dietary interventions including the improvement of insulin sensitivity [89]. The abundance of a mucin-degrading bacterium, *Akkermansia muciniphila*, is inversely associated with body fat mass and glucose intolerance, suggesting that *A. muciniphila* might be useful in dietary intervention for obesity treatment [89]. Specific signatures of the microbiota correlate with enhanced longevity in humans [90]. Commensal bacteria in the intestinal mucus layer produce a variety of metabolites through their metabolic activity and modification of the molecules derived from the host. In addition to host metabolism, the gut microbial community through its metabolic activities produces compounds that can act on the surrounding microenvironment and through circulation systemically affect other distant organs. Metabolites produced from the gut microbiome play a critical role in the maintenance of the metabolic condition and homeostasis of the host [90] and have been implicated in the regulation of age-related diseases and in enhancing longevity [90]. Probiotics are able to play a significant role in maintaining intestinal health by modulating the production of ROS and consequently the antioxidant level in the gut, suggesting that the dietary intervention with certain probiotics may be effective for diseases including cancer [91,92,93,94]. A list of intestinal microbes that exist at a high ROS state and microbes that reduce ROS load has been summarized in [95].

Data-driven dietary interventions based on the type of cancer and the microbial profile of each individual may be harnessed to promote the cancer-suppressing activities of the microbiome. Precision probiotics containing gut commensal bacteria, the benefits of which have been identified by ex vivo studies and in silico studies, might promote colonization and enhanced impacts on the host [96]. Mechanistic understanding of microbe-derived factors that impact the progression of cancer and the remedy is crucial for the establishment of “postbiotic” therapies, involving the supplementation of diet with discrete, well-defined, and well-characterized biomolecules with the microorganisms that generate these molecules [97]. Several phytochemicals with potential therapeutic value have been identified and applied for the intervention of fatty liver disease associated with metabolic dysfunction, cardio-metabolic disorders, and neurodegenerative diseases such as amyotrophic lateral sclerosis [98,99,100,101]. Several phytochemicals, such as resveratrol and anthocyanins, exhibited potential activity against metabolic-dysfunction-associated fatty liver disease [98]. Resveratrol has potential therapeutic value for amyotrophic lateral sclerosis [101]. The relation between the species diversity of the intestinal microbiota and phenotypes of diseases may be attributed to the modification in function in the mitochondrion and ROS production [102]. Although how changes in mitochondrial function and ROS production could contribute to the relationship between the species diversity of the gut microbiome and the phenotypes of common diseases remains largely unknown; one possibility is proposed that the redox status within the mitochondrion may be regulating the intestinal microbiome by the metabolites in the mitochondrion [102] (Figure 2).

## 7. Perspective

Multiple therapies to treat cancers have been developed, with the major ones being chemotherapy with anticancer drugs meant to kill cancer cells and radiation therapy used at doses high enough to kill cancer cells and shrink tumors. Other therapies include the surgical removal of cancerous tumors and targeted therapy with molecularly targeted drugs. However, resistance to chemotherapy and radiation has frequently been observed because of the intrinsic properties that cancer stem cells acquire through multiple interactions with the tumor microenvironment, including the dysregulation of the drug-efflux pump system and the DNA repair capacity. Cell plasticity is an important driver of drug resistance. Despite the cancer stem cell plasticity, which hampers the application of anti- cancer stem cell therapy, a few pharmaceutical companies have developed methodologies for eliminating the small cell population of this type. Immune checkpoint blockade therapy is a recent promising approach for cancer treatment. The therapy might provide opportunities for the targeted elimination of cancer stem cell subsets to complete regression of malignant tumors. Cancer stem cells exhibit certain characteristics to circumvent attacks by the immune system, including altered antigen expression that causes reduced recognition by the immune system. Immune checkpoint blockade approaches can be used to target cancer stem cells. One of the major breakthroughs in cancer immunotherapy was the identification of immune checkpoint molecules, cytotoxic T-lymphocyte-associated protein 4 (CTLA-4), the programmed cell death-1 (PD-1), and its ligands programmed death-ligand 1 (PD-L1) and PD-L2 tumors. CTLA-4 is constitutively expressed in regulatory T cells (Treg), one of the subsets of CD4-positive T cells represented by the master transcription factor, forkhead Box P3 (FOXP3). PD-1 molecule is expressed in activated both CD4-positive and CD8-positive T cells and also in natural killer (NK) cells, B-lymphoma cells (B cells), dendritic cells (DC), and activated monocytes. Chimeric antigen receptor T (CAR-T) cell therapy is also emerging as another strategy to target cancer stem cells. The potential limitations of these therapies targeting cancer stem cells are attributed to the heterogeneity in cancer stem cells and their ability to escape from immunotherapies. Moreover, there are difficulties in the development of CAR-based methods with less off-target toxicity to non-cancerous cells.

Specific dietary factors are a novel and promising strategy for the investigation of their relevance to cancer. Diet has various impacts on health, lifespan, disease, and cancer incidence. Dysregulated metabolism is one of the hallmarks of cancer and inspires new therapeutic strategy, and in studies comparing cancer stem cells to non-cancer stem cells, no universal metabolic patterns have emerged. Cancer stem cells and other non-cancer stem cells preferentially utilize the glycolytic pathway and the system of oxidative phosphorylation, which depend on the cancer cell type and model system used preventing the evaluation of environmental effects, and the metabolic adaptation of cancer stem cells has developed as a critical procedure in distant colonization. Therefore, the abnormal energy requirements of cancer cells including cancer stem cells during the metastatic cascade might represent an opportunity for the treatment of the diseases. Nutrients also play a pivotal role in cancer stem cells, and thus clarifying the molecular mechanisms underlying the regulation of the response of cancer stem cells to diet has recently gained much interest to pursue cancer prevention and therapy.

A growing body of evidence suggests that the supplementation of daily food with dietary phytochemicals may promote performance during physical exercise, far better through the activation of antioxidant and anti-inflammatory signaling pathways than as direct scavengers of ROS by exerting only their antioxidant activity [103,104,105,106]. Given the essential roles of ROS in basic physiological functions, future therapeutics should be designed to modulate the redox state [23,107,108,109]. Dietary phytochemicals, microbiota, and metabolites derived from them can be expected to be developed as drugs to prevent or treat chronic disorders such as inflammation, cancer, and neurodegeneration soon.

## 8. Conclusions

ROS could become both beneficial and harmful to human health in certain conditions. Too much production of ROS may play a critical role in the pathogenesis of several diseases. Considering these pleiotropic roles of ROS in cellular physiological functions, future therapeutics should be designed to modulate the redox state in tumor microenvironment. Dietary phytochemicals, microbiota, and/or their metabolites might be promising for the development of cancer therapeutics.

## Figures and Tables

**Figure 1 antioxidants-12-01056-f001:**
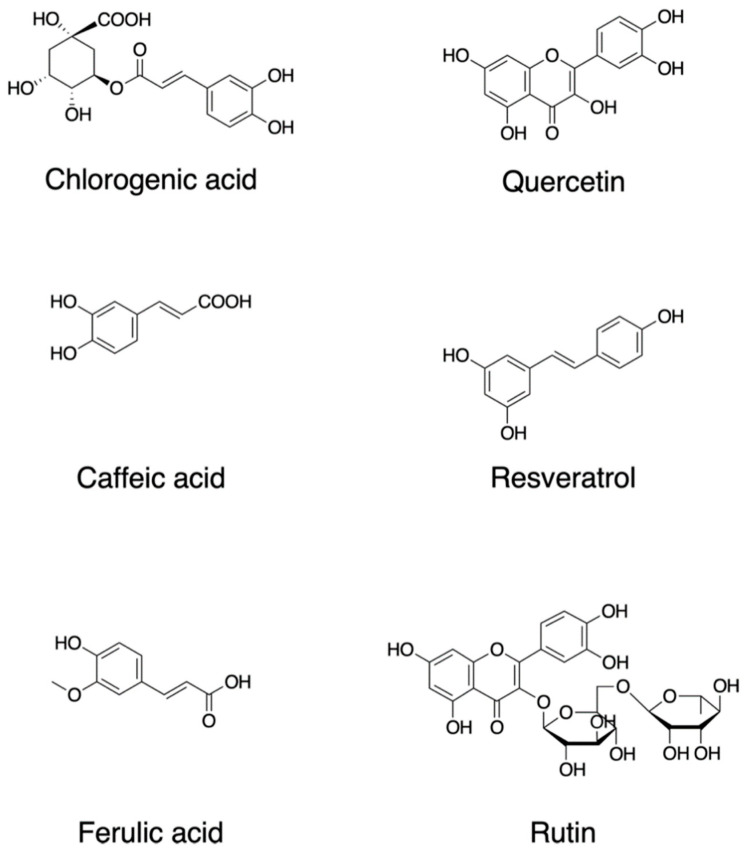
Chemical structures of dietary phytochemicals that act as antioxidants.

**Figure 2 antioxidants-12-01056-f002:**
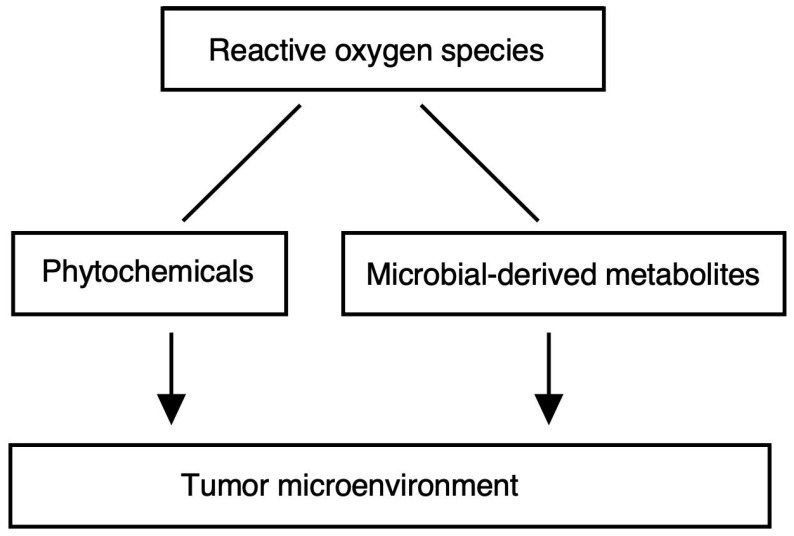
Signaling by reactive oxygen species with dietary phytochemicals and microbial-derived metabolites as potent therapeutic regulators of the tumor microenvironment.

**Table 1 antioxidants-12-01056-t001:** Reactive oxygen species (ROS): reactive species derived from molecular oxygen.

Specific Chemical Species	Chemical Formula
Free radicals	
Superoxide anion radical	O_2_^−•^
Hydroxyl radical	OH^•^
Alkoxyl radical	RO^•^
Peroxyl radical	ROO^•^
Nonfree-radical forms of ROS	
hydrogen peroxide	H_2_O_2_
ozone	O_3_
singlet molecular oxygen	O_2_^1^Δ_g_
electronically excited carbonyls	RCO
organic hydroperoxide	ROOH

## Data Availability

There are no data outside that reported in this article.

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
