# Peer review of "Pleiotropic Signaling by Reactive Oxygen Species Concerted with Dietary Phytochemicals and Microbial-Derived Metabolites as Potent Therapeutic Regulators of the Tumor Microenvironment"

_antioxidants, 2023, doi:10.3390/antiox12051056_

Round 1

Reviewer 1 Report

The manuscript entitled "Pleiotropic signaling by reactive oxygen species concerted with dietary phytochemicals and microbial derived metabolites as potent therapeutic regulators of the tumor microenvironment" represents very good approach to the effects related to the reactive oxygen species.

The article present some aspects rather rarely be analysed in respect to reactive oxygen species. It considered the role of dietary polyphenols in the ROS regulation including resveratrol, quercetin or chlorogenic acid and also the role of ROS signals in embryo development. My attention related strongly to the paragraph 4 about the signaling in the tumor microenvironment, which considered cancer stem cells (CSC) defined as small subpopulation of cells in tumor that posses the capacity to self-renew. CSCs are characterised by  their adaptality to the heterogenous tumor microenvironment. Thus, the epithelia to mesenchymal transition (EMT) is triggered by signals received by the cells from the surrounding microenvironment. However,, authors indicated that cancer cells might migrate locally without activating EMT. This phenomena is controversial and discussed in this part of work.

In my opinion, the central value of this article-review is the consideration of "multi-aspects" in the description of ROS signaling pathways and the consideration of the aspect, which was rather rarely described in ROS related works: for example the microbial profile and its imbalance in the organism.

I would like to underline that this work is based on the last literature data. In the list of references (99) more than 50 (means a half of them) were published after 2020.

My doubts relate to the page 4 line142-145 started from the "3.Results......This section may be divided......" This sentence is probably missing in this place.

Author Response

Reviewer1

the epithelia to mesenchymal transition (EMT) is triggered by signals received by the cells from the surrounding microenvironment. However, authors indicated that cancer cells might migrate locally without activating EMT. This phenomena is controversial and discussed in this part of work.

This issue was raised by Dr. David Tarin, from University of California, San Diego, at the annual AACR (American Association for Cancer Research) meeting. As in the reference #66 of the original manuscript, the great cancer biologists Dr. Robert Weinberg and Dr. Isaiah Fidler argued about this issue (Nature 2011, 472: 273. DOI: 10.1038/472273a). To avoid any confusion, we have deleted the sentences.

My doubts relate to the page 4 line142-145 started from the "3.Results......This section may be divided......" This sentence is probably missing in this place.

This sentence, "3. Results,This section may be divided by subheadings. It should provide a concise and precise description of the experimental results, their interpretation, as well as the experimental conclusions that can be drawn,” does not exist in our submitted manuscript, but may be just a kind of typo errors added during editorial process. Thus it has been removed.

Reviewer 2 Report

Report: antioxidants-2342205

The review discusses the importance of ROS in cellular regulation and signaling but also highlights the potential risks associated with their excessive generation. The authors suggest that modulating the redox state could be a useful therapeutic strategy for preventing or treating chronic disorders such as inflammation, cancer, and neurodegeneration. Although the topic is very interesting, the manuscript presents some weaknesses in the content that should be improved. Some section should be re-written and novel refs added. I have several concerns that should be addressed:

1-Line 85: The statement "The effects of these phytochemicals when plants are included in the diet on human health are largely unsolved" is not accurate. Indeed, while some effects may be unknown or require further investigation, there is a substantial body of research that has explored the health benefits of consuming plant-based foods such as polyphenols and flavonoids PMID: 15113710, 23650286, 16934632.

2-The authors state that flavonoids are potent antioxidants, which is true. However, recent research suggests that their beneficial effects on human health may not be solely due to their antioxidant properties but also to their ability to modulate signaling pathways and gene expression, such as autophagy or stress response. 28937664 This point should be included.

3-The authors mention that hydroxyl groups in flavonols are responsible for their biological activities but do not provide any examples or further explanation. It could be helpful to mention some specific biological activities that flavonoids have been shown to exert, such as anti-inflammatory, anti-cancer, and neuroprotective effects. 15113710, 26892981

4- The authors discussed the antioxidant and anti-inflammatory activities of several polyphenols, including Quercetin, Resveratrol, Rutin, and Chlorogenic acid 30827677. These compounds are found in various fruits, vegetables, and seeds and have been studied for their potential health benefits. However, more specific information about the studies that have been conducted on these compounds should be provided. For example: -it would be useful to know the doses at which these compounds were administered, the duration of the studies, and the specific health outcomes that were measured. Additionally, it would be helpful to include information about potential side effects or interactions with medications.

5- ROS as developmental signals in oocytes: Clarify the statement about the lack of ROS signals in early oocytes. While it is true that primordial oocytes in humans and Xenopus laevis lack detectable ROS signals, it is important to note that ROS levels increase as oocytes mature. This can have implications for the quality of the resulting embryos. Include these concepts in the section 11715046 17118266.

6- ROS signaling in tumor microenvironment: The authors should provide more details on the different types of cells present in the tumor microenvironment and their roles. It should be also discussed the challenges in identifying cancer stem cells based on biomarkers and the potential limitations of current therapies targeting these cells. Also, the importance of understanding the mechanisms of metastasis should be mentioned, as well as the challenges in developing therapies to prevent or treat metastasis. 33546733

7- ROS signaling pathway: The authors should provide more details on how the PI3K/AKT/PTEN axis specifically affects stem cell function, rather than jumping immediately to its role in cancer. Also, provide more clarity on how extracellular molecules modulate the PI3K/AKT/PTEN pathway and what downstream molecules are affected 34446709, 4352581. Mention that ROS signaling is linked to other signaling pathways as autophagy 31546746. Provide a sentence about therapeutic opportunity of ROS modulation.

-Line 262: The sentence "Impaired signal transduction is common in many malignancies and confers cancer cells with the advantage of enhanced proliferative and survival rates" is a bit confusing, as it implies that impaired signal transduction is beneficial for cancer cells. It would be clearer to rephrase this to something like "Mutations in components of intracellular signaling pathways can lead to dysregulated signaling in many malignancies, which can promote cancer cell proliferation and survival."

-Line 270: The sentence "Many pharmacological inhibitors of the pathway have been developed" could be improved by providing more specific information on what these inhibitors target and how they are used in cancer therapy.

- Microbiota and ROS: The authors should provide a more detailed explanation of how specific bacterial species are associated with a healthier metabolic status and how they can be used to design dietary interventions aimed at improving insulin sensitivity. Additionally, the text should be expanded to discuss the role of probiotics in greater detail, including how they maintain antioxidant levels, support the immune system, and provide barrier protection. More information should also be provided on the phytochemicals with potential therapeutic value that have been identified and applied for the treatment of various diseases. Finally, the authors should provide a more detailed explanation of how changes in mitochondrial function and ROS production could contribute to the relationship between species diversity of the gut microbiome and the phenotypes of common diseases. 22250971, 35067448 35517619

-There is only 1 figure in the manuscript: It would be helpful to include visuals, such as tables or graphs, to illustrate the concepts being discussed. This can help readers better understand the main message of each section and the relationships between different topics.

Author Response

Reviewer2

 1-Line 85: The statement "The effects of these phytochemicals when plants are included in the diet on human health are largely unsolved" is not accurate. Indeed, while some effects may be unknown or require further investigation, there is a substantial body of research that has explored the health benefits of consuming plant-based foods such as polyphenols and flavonoids PMID: 15113710, 23650286, 16934632.

We highly appreciate the variable comments, and modified the sentence (lines 87-90 of the revised manuscript) by adding a new reference (PMID: 15113710) as no.24.

In addition to this ‘15113710’, there are many 8-digit numbers (23650286, 16934632, 28937664 , 26892981, 30827677, 11715046, 17118266, 33546733, 34446709, 4352581, 22250971, 35067448, 35517619) in Reviewer2’s comments. However, we could not find the relevant papers. We would appreciate it if Reviewer2 could inform us the titles of the articles these numbers refer to.

2-The authors state that flavonoids are potent antioxidants, which is true. However, recent research suggests that their beneficial effects on human health may not be solely due to their antioxidant properties but also to their ability to modulate signaling pathways and gene expression, such as autophagy or stress response. 28937664 This point should be included.

 We thank the reviewer for the insightful comments. The issue concerning flavonoids’ action other than their antioxidant properties is addressed in the revised manuscript (L135-141).

3-The authors mention that hydroxyl groups in flavonols are responsible for their biological activities but do not provide any examples or further explanation. It could be helpful to mention some specific biological activities that flavonoids have been shown to exert, such as anti-inflammatory, anti-cancer, and neuroprotective effects. 15113710, 26892981

 We thank the reviewer for the constructive comments. According to the advice, further explanation for flavonoids’ specific biological activities has been added (L97-99, 113-117, 125-126).

4- The authors discussed the antioxidant and anti-inflammatory activities of several polyphenols, including Quercetin, Resveratrol, Rutin, and Chlorogenic acid 30827677. These compounds are found in various fruits, vegetables, and seeds and have been studied for their potential health benefits. However, more specific information about the studies that have been conducted on these compounds should be provided. For example: -it would be useful to know the doses at which these compounds were administered, the duration of the studies, and the specific health outcomes that were measured. Additionally, it would be helpful to include information about potential side effects or interactions with medications.

 We thank the reviewer for the constructive comments. According to the comments, the sentences addressing these issues have been added to the text (L159-167). The new references have been added (no.49-52 of the revised manuscript).

5- ROS as developmental signals in oocytes: Clarify the statement about the lack of ROS signals in early oocytes. While it is true that primordial oocytes in humans and Xenopus laevis lack detectable ROS signals, it is important to note that ROS levels increase as oocytes mature. This can have implications for the quality of the resulting embryos. Include these concepts in the section 11715046 17118266.

 We thank the reviewer for the insightful comments. The manuscript has been modified accordingly (L187-189). 

6- ROS signaling in tumor microenvironment: The authors should provide more details on the different types of cells present in the tumor microenvironment and their roles. It should be also discussed the challenges in identifying cancer stem cells based on biomarkers and the potential limitations of current therapies targeting these cells. Also, the importance of understanding the mechanisms of metastasis should be mentioned, as well as the challenges in developing therapies to prevent or treat metastasis. 33546733

According to the advice, the following information has been added to the revised manuscript: more details on the different types of cells present in the tumor microenvironment and their roles (L202-208, ref no.57); the challenges in identifying cancer stem cells based on biomarkers (L225-229, refs no.58 and 59); the potential limitations of current therapies targeting these cells (L377-401); the importance of understanding the mechanisms of metastasis (L273-277, refs no.72 and no.73); the challenges in developing therapies to prevent or treat metastasis (L277-279).

7- ROS signaling pathway: The authors should provide more details on how the PI3K/AKT/PTEN axis specifically affects stem cell function, rather than jumping immediately to its role in cancer. Also, provide more clarity on how extracellular molecules modulate the PI3K/AKT/PTEN pathway and what downstream molecules are affected 34446709, 4352581. Mention that ROS signaling is linked to other signaling pathways as autophagy 31546746. Provide a sentence about therapeutic opportunity of ROS modulation.

According to the advice, the following information has been added to the revised manuscript: details on how the PI3K/AKT/PTEN axis specifically affects stem cell function (L284-288, refs no.74 and no.75), how extracellular molecules modulate the PI3K/AKT/PTEN pathway and what downstream molecules are affected (L293-295refs. No.77 and no.78); ROS signaling is linked to other signaling pathways as autophagy (L320-324, ref. no.86); therapeutic opportunity of ROS modulation (L324-326, ref. no.87).

-Line 262: The sentence "Impaired signal transduction is common in many malignancies and confers cancer cells with the advantage of enhanced proliferative and survival rates" is a bit confusing, as it implies that impaired signal transduction is beneficial for cancer cells. It would be clearer to rephrase this to something like "Mutations in components of intracellular signaling pathways can lead to dysregulated signaling in many malignancies, which can promote cancer cell proliferation and survival."

 We thank the reviewer for carefully reading our manuscript. The sentence has been modified accordingly (L301-303).

-Line 270: The sentence "Many pharmacological inhibitors of the pathway have been developed" could be improved by providing more specific information on what these inhibitors target and how they are used in cancer therapy.

The more information on PI3K pharmacological inhibitors has been added (L3310-365, refs. no.81 and no.82).

- Microbiota and ROS: The authors should provide a more detailed explanation of how specific bacterial species are associated with a healthier metabolic status and how they can be used to design dietary interventions aimed at improving insulin sensitivity. Additionally, the text should be expanded to discuss the role of probiotics in greater detail, including how they maintain antioxidant levels, support the immune system, and provide barrier protection. More information should also be provided on the phytochemicals with potential therapeutic value that have been identified and applied for the treatment of various diseases. Finally, the authors should provide a more detailed explanation of how changes in mitochondrial function and ROS production could contribute to the relationship between species diversity of the gut microbiome and the phenotypes of common diseases. 22250971, 35067448 35517619

According to the comments, the information has been added to the revised manuscript: a more detailed explanation of how specific bacterial species are associated with a healthier metabolic status (L329-332, ref. no.88); how they can be used to design dietary interventions aimed at improving insulin sensitivity (L334-337); the role of probiotics in greater detail, including how they maintain antioxidant levels (L348-350); more information on the phytochemicals with potential therapeutic value (L362-365); a more detailed explanation of how changes in mitochondrial function and ROS production could contribute to the relationship between species diversity of the gut microbiome and the phenotypes of common diseases (L367-371).

-There is only 1 figure in the manuscript: It would be helpful to include visuals, such as tables or graphs, to illustrate the concepts being discussed. This can help readers better understand the main message of each section and the relationships between different topics.

Figure 2 has been added. We thank the reviewer for careful reading and allowing us significant improvement our manuscript.

Round 2

Reviewer 2 Report

I thanks the authors for taking in consideration all my suggestions. I believe that all concerns have been addressed. Overall, the manuscript was significantly improved during revision, since they included novel concepts, clarify some sentences as required and added novel relevant refs. I apologize for the mistake in digiting PMID numbers, however, I believe you included all refs correctly.

I have no more issues, the manuscript can be accepted in the present form.